# Predicting Prostate Cancer Progression During Active Surveillance Using Longitudinal bpMRI Scans and A Multi-scale Foundation Model

**Yifan Wang**[1,4]                                                        WANGYFAN@UMICH.EDU

**Bin Lou**[1]                                                      BIN.LOU@SIEMENS-HEALTHINEERS.COM

**Heinrich von Busch**[2]                      HEINRICH.VON_BUSCH@SIEMENS-HEALTHINEERS.COM

**Robert Grimm**[2]                                ROBERTGRIMM@SIEMENS-HEALTHINEERS.COM

**Sanoj Punnen**[3]                                                   S.PUNNEN@MED.MIAMI.EDU

**Dorin Comaniciu**[1]                            DORIN.COMANICIU@SIEMENS-HEALTHINEERS.COM

**Ali Kamen**[1]                                         ALI.KAMEN@SIEMENS-HEALTHINEERS.COM

**ProstateAI Clinical Collaborators**[*]

[1] *Digital Technology and Innovation, Siemens Healthineers, Princeton, NJ, USA*

[2] *Diagnostic Imaging, Siemens Healthineers AG, Erlangen, Bavaria, Germany*

[3] *University of Miami, Miller School of Medicine, Miami, FL, USA*

[4] *University of Michigan, Electrical Engineering and Computer Science, Ann Arbor, MI, USA*

**Editors:** Accepted for publication at MIDL 2025

## Abstract

Active Surveillance (AS) is the recommended management strategy for patients with low- or intermediate-risk Prostate Cancer (PCa), providing a safe alternative that helps avoid the adverse effects of overtreatment. While artificial intelligence (AI)-based models for PCa detection have been extensively studied, their application in AS remains challenging, with limited research addressing the detection of PCa progression in AS scenarios. In this study, we present a novel framework for predicting PCa progression within AS protocols using bi-parametric MRI (bpMRI). Due to the limited availability of longitudinal bpMRI scans (206 patients in our study), we first developed a multi-scale foundation model trained on a large cohort of single-year bpMRI scans, comprising 5,162 patients from 10 different institutions. Building on this foundation model, we designed a three-module framework: (1) a lesion detection module to identify PCa lesions in full bpMRI scans, (2) a lesion classification module to perform detailed analysis of the identified lesion regions, and (3) a multi-scan lesion progression prediction module to assess changes in lesions over time using longitudinal bpMRI patches. The proposed framework was evaluated on a cohort from an AS clinical trial and demonstrated significant performance improvements over baseline models and radiologists, highlighting its potential to enhance clinical decision-making in AS management.

**Keywords:** Prostate Cancer Diagnosis, Active Surveillance, Foundation Model, Longitudinal bpMRI.

---

[*] A list of members and affiliations appears at the end of the paper

## 1. Introduction

Prostate Cancer (PCa) is the second most commonly diagnosed cancer and the sixth leading cause of cancer-related deaths among men worldwide (Siegel et al., 2023). A substantial proportion of newly diagnosed cases involve patients with low- or intermediate-risk localized PCa, for whom curative treatments such as surgery or radiation offer limited benefits but carry a significant risk of adverse side effects (Michaelson et al., 2008). In clinical practice, Active Surveillance (AS) is widely recognized as a safe and effective alternative to immediate treatment (Kinsella et al., 2018; Dall'Era and Evans, 2012), aiming to minimize the overtreatment of low-risk PCa. Under AS protocols, radiologists monitor PCa progression through follow-up evaluations, including Prostate-Specific Antigen (PSA) tests and repeat prostate biopsies, with intervention considered only if progression is detected (Adamy et al., 2011; Tosoian et al., 2015). Recently, Magnetic Resonance Imaging (MRI) has gained prominence in AS protocols due to its potential to improve monitoring and reduce the need for invasive procedures such as biopsies. Nevertheless, standardized guidelines for its use in AS remain unclear (Baboudjian et al., 2022; Kinsella et al., 2018).

Currently, artificial intelligence (AI)-driven models have demonstrated significant advantages in detecting and assessing PCa using MRI scans from a single time point (Yu et al., 2020b; De Vente et al., 2020). However, relatively few studies have explored methods for detecting PCa progression in AS scenarios (Jones et al., 2021; Bozgo et al., 2024). Existing progression calculators based on clinical test results have shown moderate performance (Tomer et al., 2021; Lee et al., 2022), but they often fail to fully utilize the rich information available in MRI data. Deep learning-based approaches leveraging model-extracted features have shown improved capabilities, but most are limited to single-year MRI examinations (Sushentsev et al., 2021). For patients in AS, where the primary goal is to detect histological biopsy upgrading of a candidate lesion, a single-year MRI scan may not provide sufficient information. Some deep learning models have attempted to combine baseline and follow-up MRI exams to identify PCa progression (Sushentsev et al., 2021, 2023). However, the limited availability of datasets containing longitudinal MRI scans for individual patients often restricts these methods to small models with constrained feature spaces, focusing on patient-level diagnosis rather than identifying which specific lesion has progressed. In the deep learning domain, pretraining foundation models (Bommasani et al., 2021) on large-scale imaging datasets using self-supervised learning (He et al., 2022) has emerged as a promising solution to address the challenges posed by limited task-specific datasets and annotations (Zhou et al., 2023; Pai et al., 2024). Following pretraining, foundation models can be applied to task-specific problems, improving generalization, especially in tasks with small datasets. To the best of our knowledge, this approach has not yet been explored for detecting PCa progression in patients undergoing AS.

Building on the strengths of previous studies while addressing their limitations, we propose an end-to-end PCa progression prediction framework for AS protocols using bi-parametric MRI (bpMRI). This framework introduces several key innovations that set it apart from prior work. 1) First, we developed a multi-scale foundation model trained on a large dataset of single-year bpMRI scans, comprising 5,162 cases from 10 institutions. This model was designed to handle follow-up tasks where data availability is limited. The multi-scale design enables the model to effectively process information at both the full

bpMRI scan level and the lesion Region of Interest (ROI) level. 2) Second, we incorporated a transformer-based architecture into the framework. This architecture processed 2D inputs at the full bpMRI scan level while functioning as a sequential-3D model at the lesion level. By combining the strengths of 3D modeling for detailed lesion analysis with the computational efficiency of 2D processing, the framework minimizes processing time while maintaining high-resolution insights into lesion characteristics. 3) Third, the framework integrated longitudinal data by combining deep features extracted from lesion patches in both prior and current-year bpMRI scans. This approach enables the model to detect lesion progression over time, resulting in enhanced performance compared to methods that rely on single-year data alone. 4) Finally, our framework outperformed both clinical radiologists and a Res-UNet-based PCa diagnosis model. These results underscore its potential to advance PCa progression detection in AS protocols, providing an accurate, efficient, and clinically meaningful tool for managing patients undergoing AS.

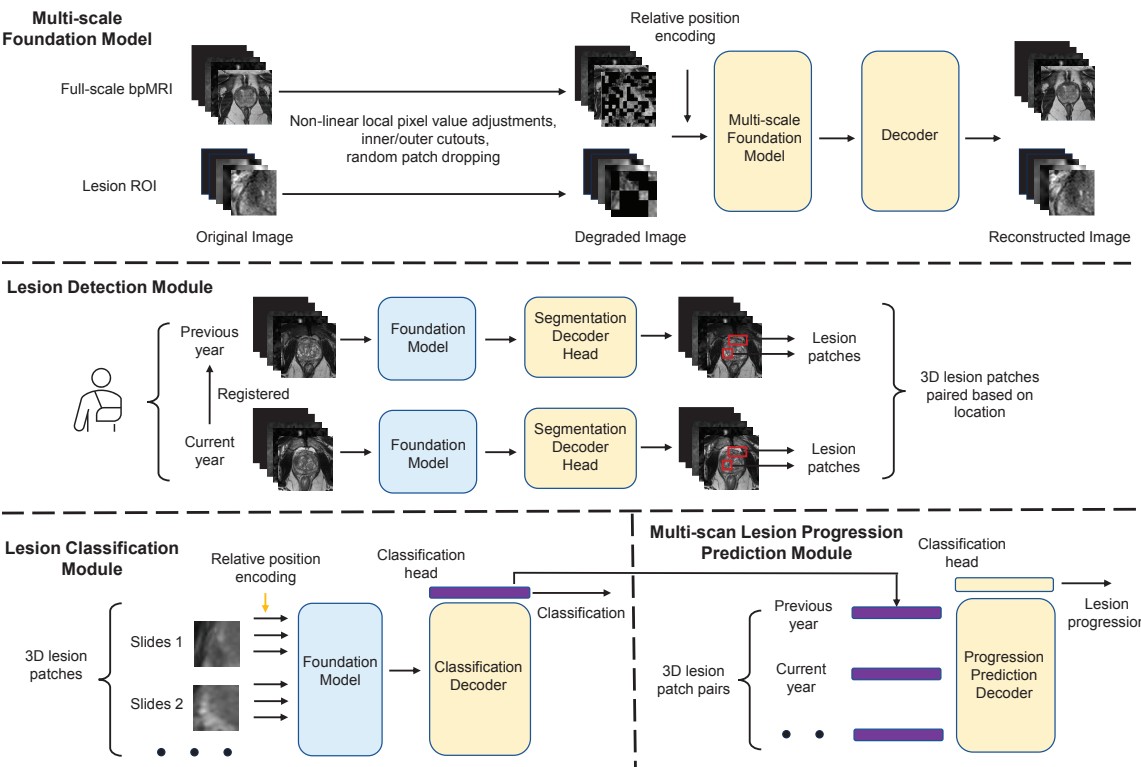

Figure 1: Schematic diagram of our PCa progression prediction framework. The yellow blocks represent the network components trained within this module, while the blue blocks indicate network components with weights from the previous step.

## 2. Methods

As illustrated in Figure 1, our PCa progression prediction framework is composed of four key modules: a multi-scale foundation model, a lesion detection module, a lesion classification module, and a multi-scan lesion progression prediction module. The multi-scale foundation

model extracts representative features from bpMRI. Based on these extracted features, the lesion detection module identifies potential lesions across the full bpMRI scans, while the lesion classification and progression prediction modules focus on analyzing each detected lesion region in greater detail.

## 2.1. Multi-scale Foundation Model

Given the unique challenges of progression prediction in AS, particularly the limited data availability, direct supervised training for progression remains constrained. In this study, while a large volume of single-year bpMRI scans was available, only a relatively small subset contained annotated PCa lesions, and longitudinal bpMRI scans were extremely limited. To address these challenges, we first developed a multi-scale foundational model using a self-supervised learning framework on the large-scale collection of single-year bpMRI scans. This foundational model was then integrated into the supervised modules of our PCa progression prediction framework.

### 2.1.1. MODEL STRUCTURE AND SELF-SUPERVISED LEARNING APPROACH

Our self-supervised model utilized an encoder-decoder architecture, where the encoder functioned as the foundational model, and the decoder assisted in training the encoder but was discarded once training was completed. For the core network block, we adopted the Vision Transformer (ViT) structure (Dosovitskiy, 2020). To tailor the model to the specifics of our problem and the available dataset, we customized a ViT-tiny architecture with 12 transformer blocks in the encoder, each containing 3 attention heads. The patch size was set to $16 \times 16$, with an output dimension of 64 for each patch.

During training, the original 2D bpMRI scans underwent a series of random transformations, including non-linear local pixel value adjustments, inner/outer cutouts (Zhou et al., 2021), and random patch dropping (He et al., 2022), to generate degraded images. This combination of local and patch-level modifications ensures the model learns meaningful and robust features through self-supervised tasks. The degraded images were processed by the ViT-tiny encoder, followed by a 3-block transformer decoder (He et al., 2022), which reconstructed the original images. The reconstruction was optimized using the Mean Squared Error (MSE) loss between the original and reconstructed images, without requiring any labels.

### 2.1.2. MULTI-SCALE INPUTS

In the PCa progression prediction framework, the models must process bpMRI scans at both the full scale to detect lesions and at the local scale to analyze lesion Regions of Interest (ROI). Both scales relied on the foundation model to effectively leverage the large volume of single-year bpMRI scans for enhanced feature extraction and analysis. To achieve this, we developed a multi-scale training method that enables the foundation model to handle variable image sizes.

Initially, the foundation model was trained on full-scale bpMRI scans ($240 \times 240$ pixels), as outlined in Section 2.1.1. Following this, the model underwent additional training on cropped bpMRI patches ($64 \times 64$ pixels) generated by the case-level lesion detection module, as detailed in Section 2.2. To support this process, we modified the positional embedding

strategy by retaining the relative positional encoding from the full scan for each patch. This adjustment provides the model with spatial context, helping it understand each patch's location relative to the entire bpMRI scan and enhancing its ability to analyze patch-level features within the broader scan.

## 2.2. Lesion Detection Module

Building upon the pre-trained foundation model, we integrated it with a transformer-based decoder (Cheng et al., 2022) as the segmentation head to detect potential lesion patches. This module was trained as a segmentation task, producing five pixel-wise binary segmentation masks corresponding to five categories: Prostate Imaging Reporting and Data System (PI-RADS) $\geq 3$, $\geq 4$, and Gleason Score Grade Group (GGG) $\geq 1$, $\geq 2$, $\geq 3$. This design enables precise lesion detection and categorization, effectively utilizing the label information available in our dataset. During training, binary cross-entropy and the Dice Similarity Coefficient were used to compute individual losses for each category-level binary segmentation mask. These losses were then combined using appropriate weighting to obtain the total training loss.

Upon completion of the training process, the output from the penultimate layer for the GGG $\geq 2$ label was used as a heatmap to propose clinically significant lesion areas. Each pixel value in the heatmap represents the probability of lesion presence at that specific location, indicating the associated malignancy risk. To extract lesion ROI patches, we identified the center of each segmented area and cropped a 3D lesion patch of size $80 \times 80 \times$ slides pixels. The number of slides for each lesion was kept flexible, varying based on the segmentation results.

## 2.3. Lesion Classification Module

Using the pre-trained foundation model, we integrated a 3-block transformer structure with an additional class token to perform lesion classification. The output consists of patch-level predictions for the five categories outlined in Section 2.2. Since our patches are 3D and may contain varying numbers of slices, we treated each slice as a token in a sequence, similar to variable-length sentences, as shown in Figure 1. During training, to address the variability in the size of each data point, we set the batch size to 1 and accumulated gradients over 16 mini-batches before updating the model's weights.

## 2.4. Multi-scan Lesion Progression Prediction Module

For longitudinal bpMRI lesion progression prediction, each 3D lesion patch was first processed by the lesion classification module. The output from the penultimate layer of the classification head is a 64-dimensional latent feature vector, representing the 3D candidate patch regardless of the number of slices it contains. Subsequently, the latent features from these patches, including lesion image patches from both the current and previous year's bpMRI scans, were passed through a one-block transformer structure with an additional class token to generate the final detection result. This design adds flexibility, allowing the framework to incorporate lesion patch information from multiple years, thus enabling the detection of lesion progression in subsequent years, such as the third or fourth year.

During training, due to the limited availability of cases with follow-up bpMRI scans, we first generated many synthetic lesion pairs from the large-scale collection of single-year bpMRI scans. Specifically, we randomly selected a malignant lesion patch (with GGG $\geq$ 2) from a patient as the current-year lesion and identified the prostate sector in which the lesion was located. Next, we selected another lesion with GGG $<$ 2 from a different patient but in the same prostate sector, referring to it as the previous-year lesion, and paired these two patches together to form a progressed synthetic lesion pair. Using a similar approach, we also selected non-malignant lesions (with GGG $<$ 2) as the current-year lesions and paired them with another non-malignant lesion from the same location in different patients to create non-progressed synthetic lesion pairs. To ensure a meaningful progression pattern, we ensured that the GGG score for the current-year lesion was always greater than or equal to that of the previous-year lesion. The module was first pre-trained using these synthetic lesion pairs. Following this, we used real lesion pairs from the active surveillance dataset, where each lesion was paired with its corresponding image patches from the previous year's bpMRI scans to detect lesion progression. To ensure the patches come from the same location, the previous year's bpMRI scans were registered to the current year's scans. A 3-fold cross-validation was employed during this step to assess the model's performance.

## 3. Experiments and Results

### 3.1. Datasets and Implementation

The data used in this study can be divided into two cohorts. The first cohort consists of 5,162 cases from 10 different institutions, each containing a single-year prostate MRI examination. In this cohort, patients undergo a single MRI exam from one specific year and receive a diagnosis. Based on this diagnosis, the patient either proceeded with treatment or was discharged, without any follow-up exams. The ground truth labels include both case-level and lesion-level annotations (with voxel-wise lesion annotations, when applicable), along with PI-RADS scores and GGG scores. The PCa lesion annotations were obtained based on the clinical radiology reports and carefully reviewed by an expert radiologist with five years of experience in radiology, specializing in prostate MRI examinations. Details of the annotation process can be found in Appendix D.

The second cohort is a smaller dataset collected as part of the Miami Active Surveillance Trial at the University of Miami. It includes 206 cases diagnosed with low to intermediate-risk PCa who chose to manage their cancer using the AS protocol. This protocol involves an initial MRI and biopsy with MRI targeting, followed by annual imaging and biopsies for the next three years or until AS endpoints are reached, resulting in a total of 458 single-year MRI scans. Consecutive MRI exams (previous year + current year) from the same cases were paired to create case pairs. For cases with three or four MRI exams, this resulted in two or three case pairs per patient. Only case pairs with radiologist-labeled lesions were included, yielding a total of 232 case pairs in this cohort. The ground truth labels for this cohort include the prostate sector locations where radiologists detected lesions, along with their respective PI-RADS scores and GGG scores from both target and systematic biopsies. There was no specific annotation step for this clinical trial cohort, and all labels were provided by clinical radiologists and derived from radiology and biopsy reports directly.

For each MRI exam, the input to our modules included bpMRI data, which comprised axial T2-weighted (T2W) acquisition, apparent diffusion coefficient (ADC) images, diffusion-weighted imaging (DWI b-2000), and binary masks of the transition zone and peripheral zone of the prostate. A detailed description of the MRI preprocessing steps can be found in Appendix C.

### 3.2. Evaluation Metrics and Baseline Models

We primarily evaluated our framework using the second cohort, which includes multi-year bpMRI scans collected under the AS protocol. Lesion-level progression was defined as a lesion that had a biopsy GGG of 1 or lower in the previous year but increased to 2 or higher in the current year. A case pair was considered positive if it contained at least one lesion with biopsy-confirmed progression. A total of 38 lesions across 38 cases met this criterion. The performance of the models was evaluated at both the case level and the lesion level. At the case level, we calculated the Area Under the Receiver Operating Characteristic Curve (AUROC), and at the lesion level, we calculated the Alternative Free-response Receiver Operating Characteristic Curve AUC (AFROC AUC). Since the ground truth does not include the exact pixel-wise location of the lesions, a true positive was defined as the presence of a detected lesion mask that overlaps with the ground truth lesion sector.

We compared our framework against radiologists and two deep learning-based baseline models. Radiologists assigned PI-RADS scores based on both the previous and current year's bpMRI scans, so the ability of PI-RADS to indicate biopsy-confirmed lesion progression reflects the radiologists' performance. For the deep learning-based baseline models, we employed a Res-UNet (Yu et al., 2020b,a) and an nnU-Net (Isensee et al., 2018), trained from scratch on the same data cohorts. Further details are provided in Appendix G.

### 3.3. Experimental Results and Model Comparison

Figure 2 presents the main results of our PCa progression detection framework. At the lesion level (left), our method achieved an AFROC AUC of 0.667, demonstrating a significant improvement over the Res-UNet model (0.599). Additionally, the PI-RADS points from radiologists fall below our curve. Out of the 38 progressed lesions, 24 were identified by radiologists using PI-RADS $\geq 3$ as a threshold and confirmed through targeted biopsy, while 14 were detected by systematic biopsy but missed by radiologists. Our framework successfully detected 12 of the 14 lesions missed by radiologists. At the case level (right), our method achieved slightly better performance (AUROC 0.727) compared to the Res-UNet model (0.720), although it remains slightly lower than radiologists' performance (0.745).

We further compared the performance of the multi-scan lesion progression prediction module with the use of only lesion patches from the current year. After incorporating multi-scan lesion patches, the lesion-level AFROC AUC improved to 0.667, up from 0.632. Figure 3 shows the lesion-level AFROC, along with two visualization examples based on T2-weighted imaging. The full bpMRI images are provided in Appendix E.

In routine clinical practice, single-year patient-level analysis is typically used to recommend biopsy testing during AS. To align with this approach, we compared the patient-level performance of our lesion detection module with baseline models. Specifically, the lesion detection module evaluated all slices in each single-year bpMRI scan to generate a 3D

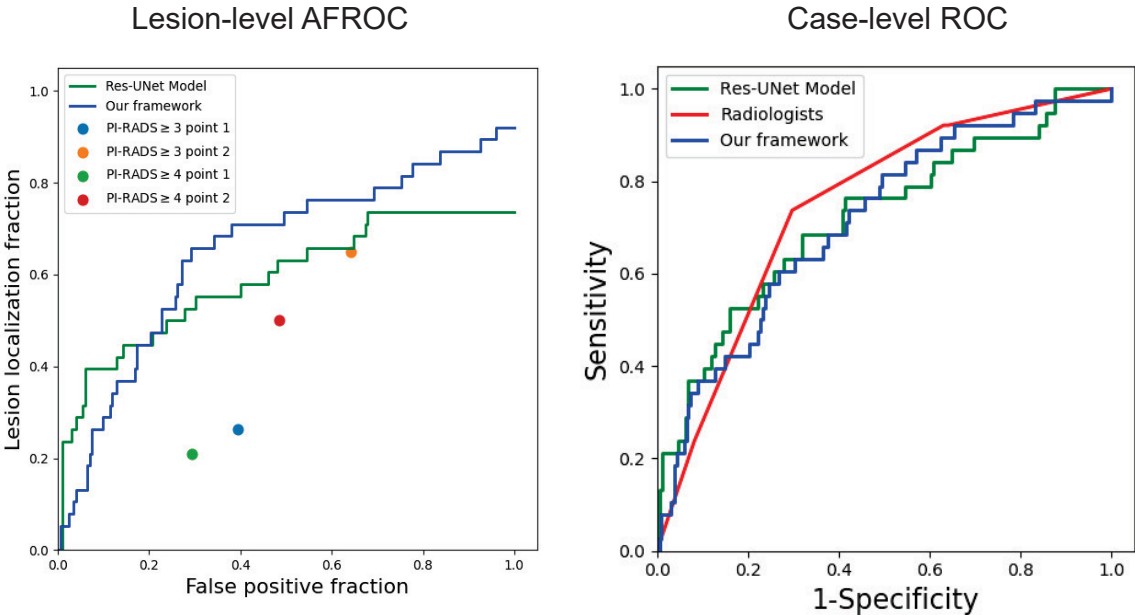

Figure 2: (Left) Lesion-level PCa progression prediction performance of our method and baseline models. "PI-RADS $\geq$ 3 point 1" represents cases where radiologists identified an increase in PI-RADS score in the current year, reaching $\geq$ 3 and "PI-RADS $\geq$ 3 point 2" reflects cases with PI-RADS scores $\geq$ 3 for the current year. "PI-RADS $\geq$ 4 point 1" and "PI-RADS $\geq$ 4 point 2" have similar definitions, only changing the threshold for PI-RADS from 3 to 4. (Right) Case-level PCa progression detection performance of our method and baseline models.

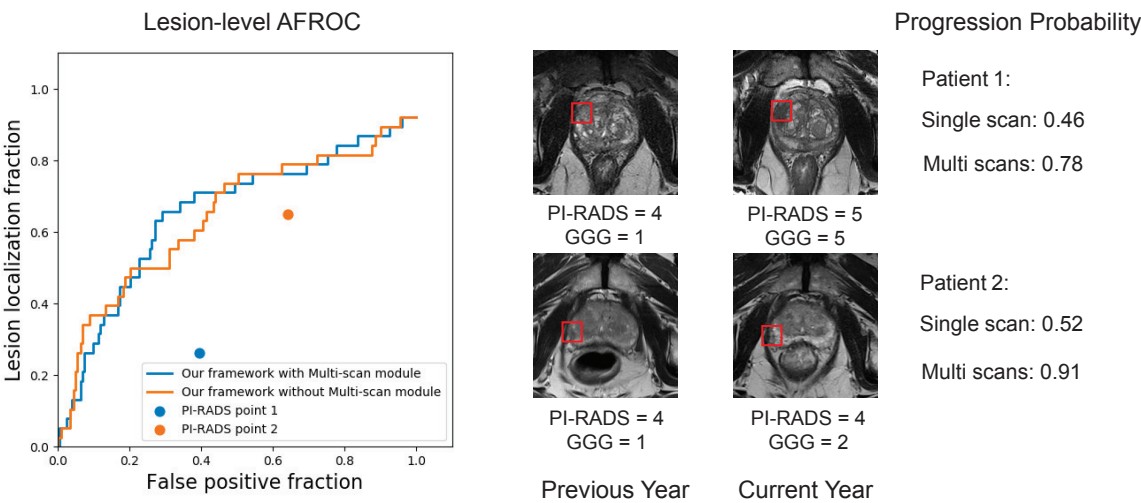

Figure 3: (Left) Lesion-level PCa progression detection performance, comparing results with and without the multi-scan lesion progression prediction module. (Right) Two T2W visualization examples where the multi-scan prediction module improves lesion progression detection.

heatmap. The maximum value from the 3D heatmap was defined as the patient's prediction score for comparison purposes. The results are shown in Table 1. We also evaluated the generalizability of our model across three additional datasets: the publicly available ProstateX-Challenge dataset (Litjens et al., 2017) and two private datasets, all of which include PI-RADS scores and biopsy results (with PI-RADS scores for the ProstateX dataset assigned by our radiology expert). Since these datasets originate from different cohorts and institutions than the training set, they serve as external validation for our model. Detailed results are provided in Appendix A. In summary, our model consistently outperformed baseline models in terms of AUROC across all datasets.

Table 1: Patient-level performance (AUROC) comparisons on AS cohort.

| Model | PI-RADS $\geq 3$ | PI-RADS $\geq 4$ | GGG $\geq 1$ | GGG $\geq 2$ | GGG $\geq 3$ |
|---|---|---|---|---|---|
| nnU-Net | 0.702 | 0.736 | 0.620 | 0.691 | 0.660 |
| Res-UNet | 0.677 | 0.734 | 0.617 | 0.734 | 0.728 |
| Radiologist | nan | nan | **0.680** | 0.738 | 0.731 |
| Our framework | **0.723** | **0.741** | 0.655 | **0.745** | **0.733** |

We also performed ablation studies to evaluate the contributions of key components of our model. The first ablation study assesses the advantages of the multi-scale design in the foundation model. Without this setting, the lesion classification module would need to be trained from scratch, resulting in a decrease in the final lesion-level AFROC AUC to 0.546. The second ablation study investigates the benefits of using a sequential-3D approach at the lesion level. When the commonly employed 2D method was applied—where the maximum score from individual 2D slices was used to determine the score for the 3D lesion—the lesion-level AFROC dropped to 0.609, highlighting the advantages of the sequential-3D approach. A summary of the results is provided in Table 2.

Table 2: Lesion-level performance (AFROC AUC) comparisons for ablation studies.

| | Full framework | w/o Multi-scan | w/o Sequential-3D | w/o Multi-scale |
|---|---|---|---|---|
| AFROC AUC | **0.667** | 0.632 | 0.609 | 0.546 |

## 4. Conclusion

This paper presents a PCa progression prediction framework for AS protocols. The approach utilizes a large cohort of single-year prostate bpMRI scans to train a multi-scale foundational model, which is then applied to task-specific modules within the framework. By incorporating longitudinal bpMRI scans, the framework significantly improves lesion progression detection compared to baseline models. It demonstrates strong potential to advance prostate cancer (PCa) progression assessment within active surveillance (AS) protocols, and exhibits promising generalizability to other diseases requiring continuous monitoring scenarios, particularly where longitudinal data are limited.

**Disclaimer** The concepts and information presented in this paper are based on research results that are not commercially available. Future commercial availability cannot be guaranteed.

**ProstateAI Clinical Collaborators** Dr. Henkjan Huisman[5], Dr. Angela Tong[6], Dr. David Winkel[7], Dr. Tobias Penzkofer[8], Dr. Ivan Shabunin[9], Dr. Moon Hyung Choi[10], Dr. Qingsong Yang[11], Dr. Dieter Szolar[12], Dr. Steven Shea[13] ,Dr. Fergus Coakley[14], Dr. Mukesh Harisinghani[15]

[5] Radboud University Medical Center; [6] New York University, New York City, NY, USA; [7] Universitätsspital Basel, Basel, Switzerland ; [8] Charité, Universitätsmedizin Berlin, Berlin, Germany ; [9] Patero Clinic, Moscow, Russia; [10] Eunpyeong St. Mary's Hospital, Catholic University of Korea, Seoul, Republic of Korea; [11] Radiology Department, Changhai Hospital of Shanghai; [12] Diagnostikum Graz Süd-West, Graz, Austria; [13] Department of Radiology, Loyola University Medical Center, Maywood, IL, USA; [14] Diagnostic Radiology, School of Medicine, Oregon Health and Science University, Portland, OR, USA; [15] Massachusetts General Hospital, Boston, MA, USA.

## Acknowledgments

We thank Nachiketh Soodana Prakash from the University of Miami, Miller School of Medicine, for his valuable assistance in organizing, collecting, and curating the dataset from the Miami Active Surveillance Trial.

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

## Appendix A. Additional External Evaluation of Our Model on Single-year bpMRIs

As an extension of Table 1, Table 3 further demonstrates the model's generalizability. Specifically, this table presents patient-level performance for single-year MRI scans across three additional datasets: the publicly available ProstateX-Challenge dataset and two private datasets, all of which include PI-RADS scores and biopsy results (with PI-RADS scores for the ProstateX dataset assigned by our radiology expert). Since these datasets originate from different cohorts and institutions than the training set, they serve as external validation for our model.

Table 3: Patient-level performance (AUROC) comparisons on three external validation set.

| Model | ProstateX (N = 343) | | | Private A (N = 170) | | | Private B (N = 181) | | |
|---|---|---|---|---|---|---|---|---|---|
| | $P \geq 3$ | $P \geq 4$ | $G \geq 2$ | $P \geq 3$ | $P \geq 4$ | $G \geq 2$ | $P \geq 3$ | $P \geq 4$ | $G \geq 2$ |
| nnU-Net | 0.840 | 0.856 | 0.808 | 0.858 | 0.841 | 0.790 | 0.846 | 0.883 | 0.770 |
| Res-UNet | **0.873** | 0.878 | 0.822 | 0.813 | 0.818 | 0.823 | 0.821 | 0.828 | 0.768 |
| Our framework | 0.861 | **0.888** | **0.832** | **0.878** | **0.891** | **0.869** | **0.852** | **0.886** | **0.824** |

Note: N represents the number of cases in the dataset, P denotes the PI-RADS scores, and G stands for the GGG scores.

## Appendix B. Training Protocols and Model Hyperparameters

### B.1. Multi-scale Foundation Model

For generating degraded images, for each image, we first selected 100 image blocks with randomly chosen lengths and widths between [1, 24] pixels and placed them at random locations. We then performed local pixel shuffling within each block. Then, with a 50% probability, the image underwent a nonlinear transformation based on Bézier Curves, as defined in (Zhou et al., 2021). In the final step, we randomly masked 25% of the $16 \times 16$ patches.

For training the multi-scale foundation model, the batch size was set to 64 for each GPU, with an initial learning rate of 0.001, using Adam as the optimization algorithm. The model was trained for 600 epochs, taking approximately 72 hours using four NVIDIA A100 GPUs.

### B.2. Lesion Detection Module

For training the lesion detection module, the batch size was set to 32 for each GPU, with an initial learning rate of 0.001, using Adam as the optimization algorithm. Early stopping was applied based on the performance on the validation set. The model was trained for around 100 epochs, taking approximately 5 hours using eight NVIDIA A100 GPUs.

After training, a threshold of 0.3 was applied to generate the segmentation areas. More details related to this choice can be found in Appendix F.

### B.3. Lesion Classification Module

For training the lesion classification module, since the input 3D image patches may have varying shapes due to different numbers of slides for each detected lesion, the batch size was set to 1 for each GPU. Gradients were accumulated over 16 mini-batches before updating the model's weights. The initial learning rate was 0.001, using Adam as the optimization algorithm. Early stopping was applied based on the performance on the validation set. The model was trained for around 50 epochs, taking approximately 5 hours using eight NVIDIA A4500 GPUs.

### B.4. Multi-scan Lesion Progression Prediction Module

During the pre-training on the synthetic lesion pairs, the same as in the lesion classification module, the batch size was set to 1 for each GPU. Gradients were accumulated over 16 mini-batches before updating the model's weights. The initial learning rate was 0.001, using Adam as the optimization algorithm. The model was trained for 100 epochs, taking approximately 8 hours using eight NVIDIA A4500 GPUs.

During the cross-validation step on the AS cohort, we selected 3 folds. The batch size was set to 1 for each GPU, and gradients were accumulated over 16 mini-batches before updating the model's weights. The initial learning rate was set to 0.0005, and Adam was used as the optimization algorithm. Early stopping was applied based on the performance on the validation set. The model was trained for 100 epochs, taking approximately 1 hour using one NVIDIA A100 GPU for each fold.

## Appendix C. Preprocessing of MRI

Initially, the original T2-weighted (T2W) and diffusion-weighted imaging (DWI) series were extracted from the raw DICOM files. We employed a voxel-wise logarithmic extrapolation of the fitted signal decay curves to compute the apparent diffusion coefficient (ADC) map and new DWI volumes at b-values of 2000 $sec/mm^2$. The ADC and DWI b-2000 images were registered to their corresponding T2W series and resampled to a voxel spacing of 0.5 mm × 0.5 mm in the axial plane while maintaining the original resolution along the slice axis. All images were normalized to facilitate the training process. T2W images were linearly normalized to the range [0, 1] based on the 0.05 and 99.5 percentiles of pixel intensity. Since ADC volumes represent quantitative parametric maps, they were normalized using a constant factor of 3000. For the DWI b-2000 volumes, we first normalized them by a factor obtained from the median intensity within the prostate gland region of the corresponding DWI b-0 volumes and then applied a constant value to linearly map the intensity range to [0, 1].

## Appendix D. Details of Annotation Process

The annotation process for Prostate Imaging Reporting and Data System (PI-RADS) scoring and voxel-level prostate lesion segmentation followed the following structured approach. First, we collected all clinical radiology reports, each containing the PI-RADS score assigned to individual lesions along with their corresponding lesion annotations. The raw clinical

annotations varied in format, including landmarks indicating the lesion, bounding boxes around the lesion, or single contours on at least one slice to guide annotators regarding the location of the lesion of interest. Next, annotators were tasked with delineating a complete 3D lesion mask based on the original annotations. All annotators received training from radiologists with Doctor of Medicine degrees and residency in Radiology. Under radiologist supervision, annotators received guidance throughout the process, addressing any uncertainties that arose. In the third step, the 3D lesion annotations from annotators underwent review and necessary corrections by radiologists within the annotation team. Radiologists possessed the authority to overrule annotators' annotations when deemed necessary. Finally, all annotations and corresponding clinical reports were meticulously reviewed by an expert radiologist with five years of experience in radiology, specializing in prostate MRI examinations. As part of the annotation, we relied on the clinical assignment of PI-RADS as determined by the original clinical site. The primary objective of the annotation process was to harmonize and extend lesion contouring across slices, providing a 3D mask for each identified lesion. The expert radiologist made minimal changes to the original PI-RADS assignment ($< 1\%$) during case reviews to ensure consistency in applying the same PI-RADS criterion across all datasets. All annotations were performed using an internally customized tool where anonymized MRI DICOM series were loaded.

## Appendix E. Visualization Examples of Multi-Scan Setting on Lesion Progression Detection

Figure 4 presents two visualization examples where the multi-scan setting notably enhances lesion progression detection performance. In both cases, lesion progression was confirmed through biopsy results, and our proposed model accurately identifies lesions that progress after one year. In the first example, the lesion exhibits an increased PI-RADS score, with the patient's predicted progression probability rising from 0.46 when using a single scan to 0.78 under the multi-scan setting. In the second example, the lesion maintains a stable PI-RADS score without any reported progression, and the predicted progression probability increases from 0.52 to 0.91 under the multi-scan setting.

## Appendix F. Threshold for Lesion Classification Module

In our experiments, we set the threshold for the output of the lesion detection module to 0.3 to generate the lesion segmentation areas.

Based on our overall pipeline design and clinical significance, we chose a relatively low threshold to achieve high sensitivity, as our subsequent lesion classification and multi-scale foundation model can help reduce false positives. This threshold was determined using the validation set during the training of the lesion detection module, ensuring that the case-level sensitivity achieved 0.95 on the validation set. Table 4 presents a sensitivity analysis for positive lesions with GGG $\geq 2$ in our AS cohort, which contains a total of 38 positive lesions, of which the radiologist identified 26 of them (PI-RADS $\geq 3$).

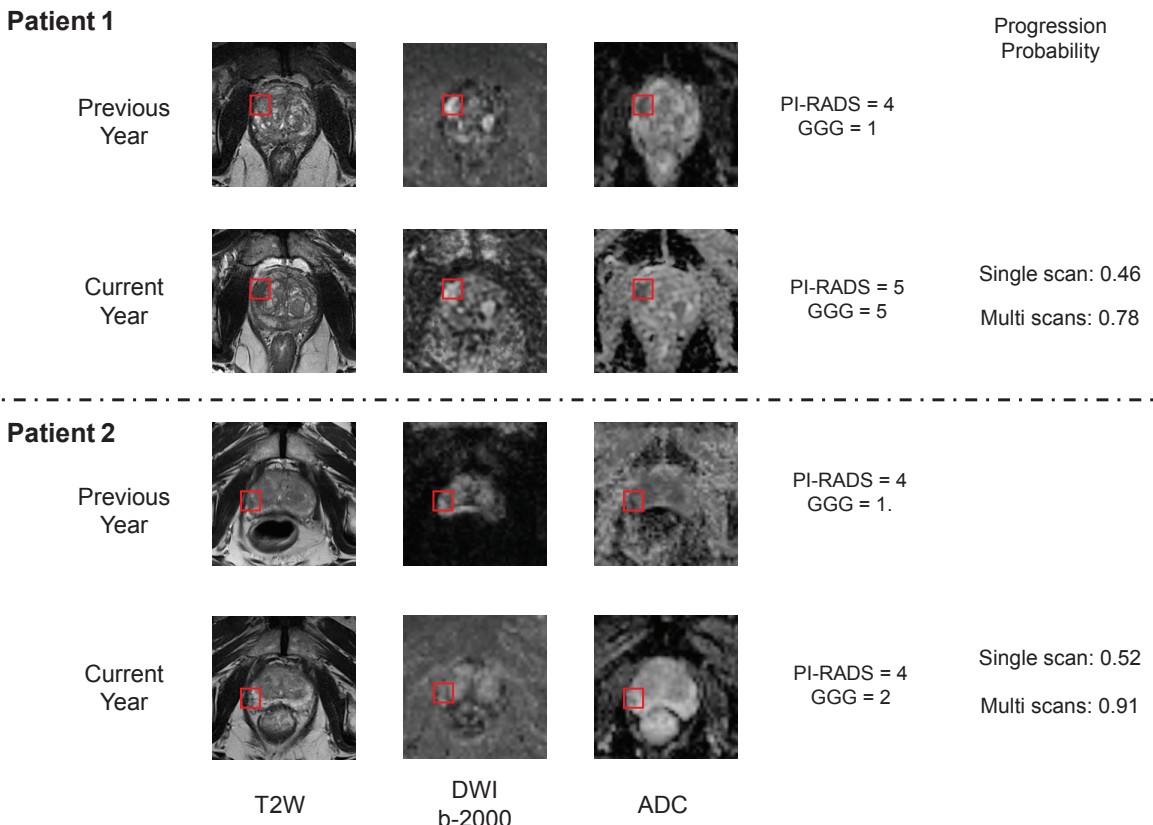

Figure 4: Two bpMRI visualization examples where the multi-scan prediction module improves lesion progression detection.

Table 4: Sensitivity analysis of the lesion detection module with varying thresholds.

| Threshold | 0.2 | 0.3 | 0.4 | 0.5 | 0.6 | 0.7 |
|---|---|---|---|---|---|---|
| Lesion-level Sensitivity | 0.921 | 0.921 | 0.868 | 0.789 | 0.658 | 0.605 |

## Appendix G. Baseline Deep-learning Models

We used the 2D Res-UNet model proposed in (Yu et al., 2020b,a), which includes a PCa detection model followed by a false-positive reduction model, and an nnU-net (Isensee et al., 2018) as the baselines for comparison. The preprocessed T2W, ADC, DWI b-2000 images, and the binary prostate mask were concatenated as input to the network. The network was trained using the first cohort, which is similar to our method. During training, we applied both binary cross-entropy (BCE) loss and Dice loss.

