# OpenReview forum: "Predicting Prostate Cancer Progression During Active Surveillance Using Longitudinal bpMRI Scans and A Multi-scale Foundation Model"
_MIDL.io/2025/Conference — MIDL 2025 Poster_

### Official Review · Reviewer_cna8 · 2025-02-16

**Confidence:** 5
**Preliminary Rating:** 3
**Final Rating:** 4

**Summary:**

In the paper, the authors investigated how AI-based models perform in PCa progression within the protocol of active surveillance, using a Transfer Learning approach, and conducted a multi-institutional research to demonstrate the investigations.

**Strengths:**

The authors’ research topic is targeting to a practical issue related to the prostate cancer detection, which is regarding the active surveillance patients. It could be clinically valuable since it shows the potential generalizability with the multi-center study and involvement of multiple clinicians.

**Weaknesses:**

Although the topic is relatively interesting and the experimental designs have some clinical meaning, the AI methodology part lacks novelty. It is a relatively general transfer learning approach when dataset size is limited. In addition, nearly no baseline comparisons with any DL models that could achieve the same functionality, or any ablation study about why and how the multi-class label design works better than the naïve designs.

**Detailed Comments:**

Please see my other sections.

**Justification Of The Final Rating:**

The authors responded my comments well, and I have no more comments to made. I have edited my rating correspondingly. The authors responded my comments well, and I have no more comments to made. I have edited my rating correspondingly.

**Justification Of The Preliminary Rating:**

The authors porposed a relatively good clinical topics, and the multi-intitutional study design showed the potential of the generalizability of the work, and thus could be clinical meaningful. However, for the AI-related part is, the novelty is limited, and the baseline comparisons are not enough, and the key ablation study is missing.

**Questions To Address In The Rebuttal:**

1. “As illustrated in Figure 1, our PCa progression prediction framework is composed of three key modules: …” you showed FOUR modules in the Figure 1. In addition, this sentence suppose to be the abstract of the following subsections, however, it does not even include the 2.1 Multi-scale foundation model. It really confuses me. Try to think like a reader and optimize the figure and your description.
2. What is the definition of “single-year bpMRI scan”? Need to give out precise definition.
3. It is very hard to state your multi-scale foundation model is a self-supervised model, since it needs to know the lesion locations and need to be labeled by the clinicians.
4. I won’t say your muti-scale foundation model is a “foundation model”, but it is a “pre-trained model”. The naming confuses people, as the foundation model is generally referred to the LLM-related topics, and refers to a way “larger” model. Your approach is a typical workflow of doing Transfer Learning. I would like to request the authors to change the naming.
5. “producing five pixel-wise segmentation masks corresponding to five categories: Prostate Imaging–Reporting and Data System (PIRADS) ≥ 3, ≥ 4, and Gleason Score Grade Group (GGG) ≥ 1, ≥ 2, ≥ 3.” How the label of each pixel was designed need to be clear. Does each pixel have a label pattern like [0, 1, 0, 1, 1]?
6. “we first generated many synthetic lesion pairs. Specifically, we cropped clinically significant lesion pairs from single-year bpMRI scans and paired them with lesions from the same location but from different patients” I would like to learn more about how did the authors do the “pair” operation.
7. I would like to learn how PI-RADS>=4 works in figure 2. It is very well-known that “if PI-RADS==3 should be treated as MRI positive” is still under lots of discussion. Since the authors have the PI-RADS>=4 map, it should be doable within the rebuttal time period.
8. Experimentations & ablation study needed to compare the multi-class label design and the binary label design to show the reason why you did the multi-class label design.

---

> ### Author Response · Authors · 2025-03-07
>
> __General response to weakness:__
>
> We trained a nnU-Net and included the related results in our revised manuscript. As the nnU-Net is a widely used and well-recognized network architecture, it serves as a strong baseline for comparison.  We also added more results as an extension of Table 1, presenting additional patient-level results based on single-year bpMRI scans across multiple datasets and institutions.
>
> __Questions To Address In The Rebuttal:__
>
> > Q1: As illustrated in Figure 1, our PCa progression prediction framework is composed of three key modules: …” you showed FOUR modules in the Figure 1. In addition, this sentence suppose to be the abstract of the following subsections, however, it does not even include the 2.1 Multi-scale foundation model. It really confuses me. Try to think like a reader and optimize the figure and your description.
>
> We apologize for the confusion here and this is a typo. There should be four key modules, as shown in Figure 1. We have revised this paragraph and carefully reviewed the rest of the revised manuscript to improve clarity for readers.
>
> > Q2:  What is the definition of “single-year bpMRI scan”? Need to give out precise definition.
>
> In the single-year bpMRI setting, patients undergo an MRI scan and receive a diagnosis. Based on this scan, the patient either proceeds with treatment or is discharged, without any follow-up scans. This makes the task a typical classification task, as we are evaluating prostate imaging findings based solely on a single snapshot from one specific year, rather than tracking changes over multiple years. We add those descriptions in the Section 3.1 of revised manuscript.
>
> For the longitudinal bpMRI setting, patients undergo multiple MRI scans, and we can pair these scans for each patient to analyze temporal changes and disease progression.
>
> >Q3: It is very hard to state your multi-scale foundation model is a self-supervised model, since it needs to know the lesion locations and need to be labeled by the clinicians.
>
> We apologize for any confusion. For our 'multi-scale foundation model,' we did not use any labels. This step was designed as a reconstruction task, where bpMRI scans undergo a series of random transformations as input, and the model aims to reconstruct the original scans, similar to a masked autoencoder. As such, no labels were required. We add some details in the Section of 2.2 of revised manuscript.

---

> > ### Author Response · Authors · 2025-03-07
> >
> > > Q4: I won’t say your muti-scale foundation model is a “foundation model”, but it is a “pre-trained model”. The naming confuses people, as the foundation model is generally referred to the LLM-related topics, and refers to a way “larger” model. Your approach is a typical workflow of doing Transfer Learning. I would like to request the authors to change the naming.
> >
> > We appreciate the reviewer’s feedback. While the term "foundation model" was initially associated with LLMs, its definition has since broadened. As described in [1], a foundation model is "a large AI model trained on a vast quantity of unlabeled data at scale, resulting in a model that can be adapted to a wide range of downstream tasks." This concept has now extended beyond natural language processing and is increasingly applied in computer vision and medical imaging. Several studies in medical imaging [2,3,4] have developed models trained on a single imaging modality and have referred to them as foundation models, aligning with the role of our multi-scale foundation model.
> >
> > Moreover, our multi-scale foundation model differs from conventional pre-trained models used in transfer learning. In typical fine-tuning workflows, a pre-trained model still requires substantial labeled data for fine-tuning while mitigating overfitting. In contrast, our foundation model remains fixed after self-supervised learning and is not fine-tuned for specific downstream tasks. Instead, we leverage the rich representations learned by the foundation model, using them as inputs for task-specific decoders. For each downstream task, the foundation model extracts features from bpMRI scans, and only a relatively small decoder is trained for that task.
> >
> > During development, we also explored applying our foundation model for prostate segmentation using a segmentation decoder, as well as for detecting other prostate-related diseases with a classification head. However, since this manuscript focuses on leveraging the multi-scale foundation model for PCa progression prediction, rather than its broader generalizability, these aspects fall beyond the scope of this work. In this study, we present its application in two key tasks: diagnosis using single bpMRI scans and progression prediction using multiple scans. Nonetheless, one of our long-term goals is to extend the multi-scale foundation model to additional downstream applications in the future.
> >
> >
> > [1] Bommasani, Rishi, et al. "On the opportunities and risks of foundation models." arXiv preprint arXiv:2108.07258 (2021).
> >
> > [2] Zhou, Yukun, et al. "A foundation model for generalizable disease detection from retinal images." Nature 622.7981 (2023): 156-163.
> >
> > [3] Xu, Hanwen, et al. "A whole-slide foundation model for digital pathology from real-world data." Nature 630.8015 (2024): 181-188.
> >
> > [4] Pai, Suraj, et al. "Foundation model for cancer imaging biomarkers." Nature machine intelligence 6.3 (2024): 354-367.
> >
> > > Q5: producing five pixel-wise segmentation masks corresponding to five categories: Prostate Imaging–Reporting and Data System (PIRADS) $\geq$ 3, $\geq$ 4, and Gleason Score Grade Group (GGG) $\geq$ 1, $\geq$ 2, $\geq$ 3.” How the label of each pixel was designed need to be clear. Does each pixel have a label pattern like [0, 1, 0, 1, 1]?
> >
> > Yes, that's correct. For the lesion detection module, the ground truth (GT) label has dimensions $[5 \times 30 \times 240 \times 240]$, where the first channel $[30 \times 240 \times 240]$ represents a binary mask of pixel-wise annotations for PI-RADS $\geq 3$; The second channel represents a binary mask for PI-RADS $\geq 4$; The third channel corresponds to GGG $\geq 1$, and so on for the remaining channels. During loss computation, binary cross-entropy and the Dice Similarity Coefficient are used to calculate individual losses for each binary mask (channel). These losses are then summed, with appropriate weighting, to obtain the total training loss.
> >
> > This design allows for flexible outputs tailored to different clinical needs. Radiologists from different clinical sites may have varying preferences: some prioritize PI-RADS $\geq 4$ for clinically significant lesions, while others focus on GGG $\geq 1$ to detect any type of cancer. We added those details in the Section 2.2 of revised manuscript.
> >
> > From a pixel-level perspective, each pixel is assigned a label pattern such as [0, 1, 0, 1, 1], indicating its classification across different thresholds. If a case contains only PI-RADS labels, its pattern might be [0, 1, -1, -1, -1], where -1 denotes missing data. These missing values are excluded from loss computation to ensure robust training.

---

> > ### Author Response · Authors · 2025-03-07
> >
> > >Q6: We first generated many synthetic lesion pairs. Specifically, we cropped clinically significant lesion pairs from single-year bpMRI scans and paired them with lesions from the same location but from different patients” I would like to learn more about how did the authors do the “pair” operation.
> >
> > First, to clarify, we used single-year bpMRI scans to generate synthetic lesion pairs. From the training set, where lesion locations and labels are known, we first randomly selected a malignant lesion patch (with GGG $\geq$ 2) from a patient as the current-year lesion and identified the prostate sector in which the lesion was located. Next, we selected another lesion with GGG $<$ 2 from a different patient but in the same prostate sector, referring to it as the previous-year lesion, and paired these two patches together to form a progressed synthetic lesion pair. Using a similar approach, we also selected non-malignant lesions (with GGG $<$ 2) as the current-year lesions and paired them with another non-malignant lesion from the same location in different patients to create non-progressed synthetic lesion pairs. To ensure a meaningful progression pattern, we ensured that the GGG score for the current-year lesion was always greater than or equal to that of the previous-year lesion. We added those descriptions in the Section 2.4 of revised manuscript
> >
> >
> > > Q7: I would like to learn how PI-RADS>=4 works in Figure 2. It is very well-known that “if PI-RADS==3 should be treated as MRI positive” is still under lots of discussion. Since the authors have the PI-RADS>=4 map, it should be doable within the rebuttal time period.
> >
> > Thank you for raising this concern. We added PIRADS $\geq$ 4 points in Figure 2 of revised manuscript. In the lesion-level AFROC analysis, a higher PI-RADS threshold results in a lower false positive fraction and lesion localization fraction. For point 1, when the PI-RADS threshold is set to $\geq$ 3, the lesion localization fraction is 0.263 and the false positive fraction is 0.395. When the threshold is changed to $\geq$ 4, the lesion localization fraction drops to 0.210, and the false positive fraction decreases to 0.294, respectively. For point 2, when the PI-RADS threshold is set to $\geq$ 3, the lesion localization fraction is 0.650 and the false positive fraction is 0.641. When the threshold is changed to $\geq$ 4, the lesion localization fraction drops to 0.500, and the false positive fraction decreases to 0.485, respectively. At the same time, those points still fall below our framework's curve.
> >
> > >Q8: Experimentations, ablation study needed to compare the multi-class label design and the binary label design to show the reason why you did the multi-class label design.
> >
> > We chose our label design for several reasons, starting with clinical needs. Our multi-class approach enables the model to achieve patient-level diagnostic performance by detecting lesions across different PI-RADS and GGG levels, as shown in Table 1 in the results part. This design offers greater flexibility, allowing the model to address a wider range of clinical tasks.
> >
> > The second reason stems from we want to effectively utilize the available label information in the dataset. In some cases, only PI-RADS labels are available, while in others, only GGG labels are provided. With our label design and loss calculation method, as described in response to your Question 5, we can calculate the loss based solely on the PI-RADS output channel and corresponding masks when only PI-RADS labels are available, while still preserving the model’s ability to generate GGG segmentation results. Additionally, for different tasks or areas of focus, we can assign varying weights to each mask channel, which also serves as a potential solution to address class imbalance.
> >
> > We agree that this is an important and intriguing topic that warrants further investigation. We plan to explore this in future work. However, we believe that this aspect is beyond the scope of our current manuscript, as our study primarily focuses on leveraging the multi-scale foundation model and longitudinal bpMRI scans for prostate cancer progression prediction.

---

> ### Comment · Area_Chair_XehW · 2025-03-17
> **Rebuttal Response: Reviewer cna8**
>
> Dear reviewer cna8,
>
> Could you please review the author's rebuttal and submit your final response for the rebuttal. It's very important to have your final review and final decision.
>
> Best, AC

---

> > ### Comment · Reviewer_cna8 · 2025-03-17
> >
> > Dear AC,
> >
> > I cannot edit my official revision comments now. But I do have changed my evaluation based on the authors' responses, from 3 to 4.
> >
> > Best,
> > Haoxin

---

### Official Review · Reviewer_H94Q · 2025-02-17

**Confidence:** 4
**Preliminary Rating:** 4
**Recommendation:** Oral
**Final Rating:** 4

**Summary:**

This paper presents a novel framework for predicting prostate cancer progression during active surveillance using longitudinal bpMRI scans. ​The framework includes a multi-scale foundation model trained on a large dataset of single-year bpMRI scans and incorporates three key modules: lesion detection, lesion classification, and multi-scan lesion progression prediction. Ablation studies were conducted to understand the impact of each part of the model (multi-scales, multi-scans, and sequential 3D ​approach for the lesion ROI. The proposed method demonstrated performance improvements over baseline models and radiologists, highlighting its relevance for active surveillance of PCa.

**Strengths:**

- Foundational model for longitudinal studies: The framework leverages a large dataset of pMRI scans from a single year to train a foundational model that can be used for longitudinal studies. It enabled the authors to bypass the limitation of access to a large cohort of longitudinal studies, which is very difficult to obtain.
- Multiscale approach: By enabling the model to examine both the full image and the ROI of the lesion, training the foundation model enables it to improve its feature extraction capability.
- Longitudinal data integration: Combines deep features from pMRI scans from the previous and current year, enabling the detection of lesion progression over time. It is flexible in terms of how many timepoints can be included. It also aligns with what is actually done by radiologists in clinical practice.
- The article is well written, organized and the figures are clear.

**Weaknesses:**

- Missing the precision of inter and intra annotator variability in the evaluation dataset.
- The longitudinal data set is collected from a single institution, which may introduce a bias. However, the difficulty of acquiring multi-institutional longitudinal data must be taken into account, and training on a multi-institutional dataset is appreciated to make the model more robust.
- The authors might consider adding an evaluation on a public data set. The PI-CAI dataset can be used for such an evaluation.
- Reproducibility: neither the code nor the dataset are made available, which makes it more hard to reproduce the presented results. The authors, however, give detailed descriptions of the model's architecture and preprocessing steps.

**Detailed Comments:**

Adding an evaluation on a public dataset could be further beneficial to the paper. Moreover, assessing the variability between annotators might also help better understand the poor performance of radiologists for "PI-RADS point 1” lesions.

**Justification Of The Final Rating:**

I would like to thank the authors for providing more comprehensive details during the rebuttal period. Specifically, I would like to thank them for adding an evaluation of the external dataset ProstateX. This helps better understand the performance of the method on datasets from external institutions.

**Justification Of The Preliminary Rating:**

The paper presents a framework for predicting PCa progression in AS using longitudinal bpMRI, leveraging a multi-scale foundation model and a well-structured multi-step approach. It effectively addresses the challenge of limited longitudinal data, enhances feature extraction, and aligns with clinical practice. Further precisions might make the paper more complete, such as detailing inter- and intra-annotator variability, and adding an evaluation on a public dataset like PI-CAI.

**Questions To Address In The Rebuttal:**

- Have you considered evaluating your approach on the PI-CAI public dataset, or is it included in your training dataset? Even if it's a single point in time, it helps to show how the model works on a different institutional dataset.
- Can you add more details about the annotations (variability of annotators, number of annotators, level of experience, etc.)?

---

> ### Author Response · Authors · 2025-03-07
>
> __General response to weakness:__
>
> We completely agree that ensuring the model's robustness and generalizability requires rigorous validation with large prospective external datasets before clinical translation. We added more results as an extension of Table 1, presenting additional patient-level results based on single-year bpMRI scans across multiple datasets and institutions. Please see our responses to your questions below.
>
> However, as one of the key motivations for this research, acquiring multi-institutional longitudinal MRI data remains highly challenging, and to our knowledge, no public dataset meets our requirements. Currently, we have access only to this AS clinical cohort. We are actively working on collecting new internal and external independent datasets, particularly longitudinal datasets from different institutions, to further validate our model's performance and enhance its generalizability in the future.
>
> We have added additional details in the appendix of the revised manuscript. However, we are unable to share the source code, as certain components of the algorithm are implemented in a commercial product. Nevertheless, we are committed to providing as much detail as possible about the model and training process and are happy to offer further information upon request.
>
> __Answers to Questions To Address In The Rebuttal:__
>
> >Q1: Have you considered evaluating your approach on the PI-CAI public dataset, or is it included in your training dataset? Even if it's a single point in time, it helps to show how the model works on a different institutional dataset.
>
> Thank you for your suggestions. Unfortunately, we are unable to directly evaluate our model on the PI-CAI dataset due to differences in preprocessing requirements. As detailed in the appendix, our preprocessing pipeline relies on two raw DWI acquisitions with different b-values to recompute ADC and B2000 images. However, the PI-CAI challenge dataset provides only precomputed ADC and high b-value images without accurate b-values for the latter, making it impossible to reconstruct the raw DWI needed for our B2000 computation.
>
> Fortunately, we identified that the ProstateX dataset is a subset of the PI-CAI dataset and includes the necessary raw DICOM files, allowing us to apply our preprocessing pipeline. We have included evaluation results of our model’s patient-level performance on single-year bpMRI scans from the ProstateX dataset, along with two additional private datasets. Since these datasets originate from different cohorts and institutions than the training set, they serve as external validation for our model. We added these descriptions in Section 3.3 and detailed results in Appendix Section A of the revised manuscript.
>
>
> | Model       | ProstateX   |   (N = 343)|           | Private A  |  (N = 170) |            | Private B | (N = 181)|            |
> |---------------|--------|--------|--------|--------|--------|--------|--------|--------|--------|
> |                  | P ≥ 3  | P ≥ 4  | G ≥ 2  | P ≥ 3  | P ≥ 4  | G ≥ 2  | P ≥ 3  | P ≥ 4  | G ≥ 2  |
> | nnU-Net           | 0.840         | 0.856      | 0.808       | 0.858  | 0.841  | 0.790  | 0.846  | 0.883  | 0.770  |
> | Res-UNet         | **0.873**   | 0.878      | 0.822       | 0.813  | 0.818  | 0.823  | 0.821  | 0.828  | 0.768  |
> | Our framework | 0.861        | **0.888** | **0.832**  | **0.878** | **0.891** | **0.869** | **0.852** | **0.886** | **0.824** |
>
> **Note:** N represents the number of cases in the dataset, P denotes the PI-RADS scores, and G stands for the GGG scores.

---

> > ### Author Response · Authors · 2025-03-08
> >
> > > Q2: Can you add more details about the annotations (variability of annotators, number of annotators, level of experience, etc.)?
> >
> > For single-year bpMRI scans, we added a detailed Annotation Process in the Appendix Section C of the revised manuscript. In summary, the PCa lesion annotations were obtained based on the clinical radiology reports and carefully reviewed by an expert radiologist with five years of experience in radiology, specializing in prostate MRI examinations.
> >
> > For the AS cohort from the University of Miami, which directly comes from the Miami Active Surveillance Trial, there was no specially designated annotation step. All results, including the PI-RADS scores and GGG scores, were provided by clinical radiologists and derived from radiology and biopsy reports. We added these descriptions in Section 3.1 of the revised manuscript.
> >
> > You mentioned 'assessing the variability between annotators might also help better understand the poor performance of radiologists for "PI-RADS point 1” lesions' in the Detailed comments. We want to clarify that this is not due to poor performance by some radiologists, but rather a limitation of the PI-RADS system itself. Specifically, the increase in GGG score may not directly correlate with an increase in PI-RADS score, and vice versa. For PI-RADS point 1, only when radiologists assign an increased PI-RADS score between two years can it be treated as an indication of lesion progression, which leads to lower sensitivity. In contrast, for PI-RADS point 2, if radiologists consistently assign a high PI-RADS value ($\geq 3$), it also indicates progression, which results in much higher sensitivity.

---

> ### Comment · Area_Chair_XehW · 2025-03-17
> **Rebuttal Response: Reviewer H94Q**
>
> Dear reviewer H94Q,
>
> Could you please review the author's rebuttal and submit your final response for the rebuttal. It's very important to have your final review and final decision.
>
> Best, AC

---

### Official Review · Reviewer_namb · 2025-02-21

**Confidence:** 4
**Preliminary Rating:** 4
**Final Rating:** 4

**Summary:**

The manuscript proposes a deep learning framework aimed at predicting prostate cancer (PCa) progression in patients under active surveillance. The authors leverage a multi-scale foundation model. Initially, the model was pre-trained on a large cohort of single-year bpMRI scans, and then it is fine-tuned using a limited longitudinal dataset. The framework integrates three modules: lesion detection, lesion classification, and multi-scan lesion progression prediction. Experimental results demonstrate that the proposed approach outperforms a Res-UNet baseline and compares favorably against radiologist assessments.

**Strengths:**

The authors proposed a multi-scale foundation model, pre-trained with self-supervised learning and adapted for both full-scan and patch-level analysis. This approach addresses data scarcity in longitudinal settings while effectively leveraging large-scale single-year data.

The manuscript provides both lesion-level and case-level performance metrics and includes ablation studies. Experimental results show the contribution of key components of the proposed model.

By comparing the model’s performance with radiologists and a state-of-the-art CNN-based baseline, the work demonstrates its potential on clinical decision-making.

**Weaknesses:**

The current evaluation is limited to one dataset. Without an external validation, the generalizability of the model is in question.

The authors can improve reproducibility, if they provide details on training parameters (e.g., learning rates, number of epochs, batch sizes for different modules) and computational resources. A threshold of 0.3 is applied for segmentation, but more insight into the selection process would be helpful.

**Detailed Comments:**

The authors propose to generate synthetic lesion pairs. It would be helpful if they discuss how these pairs mimic real longitudinal progression and discuss any potential biases introduced by this strategy.

It would be helpful if the authors can provide how the training parameters are chosen and how the segmentation threshold is determined.

**Justification Of The Final Rating:**

Thank you for addressing some of my earlier concerns with additional evidence.
I have maintained my original rating.

**Justification Of The Preliminary Rating:**

The authors proposed a multi-scale foundation model. The manuscript provides both lesion-level and case-level performance metrics and includes ablation studies. By comparing the model’s performance with radiologists and a state-of-the-art CNN-based baseline, the work demonstrates its potential on clinical decision-making.

**Questions To Address In The Rebuttal:**

Could you provide more details on the process of generating synthetic lesion pairs? How representative are these synthetic pairs compared to true longitudinal pairs in terms of imaging characteristics and progression patterns? Do you have quantitative or qualitative analyses that validate the effectiveness of this approach?

What was the rationale behind choosing a threshold of 0.3 for the lesion detection module? Was this threshold optimized on a validation set? Could you provide a sensitivity analysis showing how variations in this threshold affect performance?

Can you elaborate on the training protocols and hyperparameters used (e.g., learning rates, batch sizes, number of epochs, and computational resources)?

Are there external validation or additional experiments to demonstrate the generalizability of the model?

---

> ### Author Response · Authors · 2025-03-07
>
> __General response to weakness:__
>
> We completely agree that ensuring the model's robustness and generalizability requires rigorous validation with large prospective external datasets before clinical translation. We added more results as an extension of Table 1, presenting additional patient-level results based on single-year bpMRI scans across multiple datasets and institutions. Please see our responses to your questions below.
>
> However, as one of the key motivations for this research, acquiring multi-institutional longitudinal MRI data remains highly challenging, and to our knowledge, no public dataset meets our requirements. Currently, we have access only to this AS clinical cohort. We are actively working on collecting new internal and external independent datasets, particularly longitudinal datasets from different institutions, to further validate our model's performance and enhance its generalizability in the future.
>
> We have added additional details in the appendix of the revised manuscript. However, we are unable to share the source code, as certain components of the algorithm are implemented in a commercial product. Nevertheless, we are committed to providing as much detail as possible about the model and training process and are happy to offer further information upon request.
>
> __Answers to Questions To Address In The Rebuttal:__
>
> > Q1: Could you provide more details on the process of generating synthetic lesion pairs? How representative are these synthetic pairs compared to true longitudinal pairs in terms of imaging characteristics and progression patterns? Do you have quantitative or qualitative analyses that validate the effectiveness of this approach?
>
> First, to clarify, we used single-year bpMRI scans to generate synthetic lesion pairs. From the training set, where lesion locations and labels are known, we first randomly selected a malignant lesion patch (with GGG $\geq$ 2) from a patient as the current-year lesion and identified the prostate sector in which the lesion was located. Next, we selected another lesion with GGG $<$ 2 from a different patient but in the same prostate sector, referring to it as the previous-year lesion, and paired these two patches together to form a progressed synthetic lesion pair. Using a similar approach, we also selected non-malignant lesions (with GGG $<$ 2) as the current-year lesions and paired them with another non-malignant lesion from the same location in different patients to create non-progressed synthetic lesion pairs. To ensure a meaningful progression pattern, we ensured that the GGG score for the current-year lesion was always greater than or equal to that of the previous-year lesion. We added these details in Section 2.4 of the revised manuscript.
>
> During the training of the multi-scan lesion progression prediction module, we found that due to the limited availability of longitudinal bpMRI scans, directly applying cross-validation on the AS cohort led to network instability and convergence issues, making it difficult to obtain stable results. To mitigate this, we designed the synthetic lesion pair generation step, which serves as useful pre-training materials to facilitate the training process. At the same time, we need to mention that these synthetic pairs exhibit a gap in accurately representing true longitudinal pairs in terms of progression patterns, meaning they cannot directly replace real longitudinal pairs. We conducted an ablation study where we trained our multi-scan lesion progression prediction module solely using the generated synthetic lesion pairs, without performing cross-validation on the AS cohort. This approach resulted in a lesion-level AFROC AUC of 0.606 on the AS cohort, which is higher than the Res-UNet-based baseline model but lower than the 0.667 achieved by our proposed full model.
>
> We think a potential reason for this is that these synthetic pairs provide useful information related to malignancy and changes in malignancy risk between two scans, making them beneficial for pre-training the multi-scan lesion progression prediction module. However, the characteristics of lesions from different patients may not be truly related, as lesions can vary significantly even within the same prostate sector. Patient-specific factors, such as differences in prostate shape or structure, can make it challenging for the model to distinguish true disease progression from unrelated variations. As part of our future work, we aim to refine the synthetic pairing process by ensuring that lesions in each pair share more similar characteristics, such as internal structure or attenuation, to better capture meaningful progression patterns.

---

> > ### Author Response · Authors · 2025-03-07
> >
> > > Q2. What was the rationale behind choosing a threshold of 0.3 for the lesion detection module? Was this threshold optimized on a validation set? Could you provide a sensitivity analysis showing how variations in this threshold affect performance?
> >
> > First, we apologize for placing this information in the methods section, which may have caused confusion. The threshold was selected based on our validation set during the experimental phase. To improve clarity, we moved this information to the Experiment section.
> >
> > Based on our overall pipeline design and clinical significance, we chose a relatively low threshold to achieve high sensitivity, as our subsequent lesion classification and multi-scale foundation model can help reduce false positives. This threshold was determined using the validation set during the training of the lesion detection module, ensuring that the case-level sensitivity achieved 0.95 on the validation set. We included a table presenting a sensitivity analysis for positive lesions with GGG $\geq$ 2 in our AS cohort, which contains a total of 38 positive lesions, of which the radiologist identified 24 of them (PI-RADS $\geq 3$).
> >
> > | Threshold                   | 0.2   | 0.3   | 0.4   | 0.5   | 0.6   | 0.7   |
> > |-----------------------------|-------|-------|-------|-------|-------|-------|
> > | Lesion-level Sensitivity    | 0.921 | 0.921 | 0.868 | 0.789 | 0.658 | 0.605 |
> >
> > This sensitivity analysis demonstrated that the chosen threshold (0.3) also performs as expected on the test set, maintaining high sensitivity. We added these explanations and results in Appendix Section F of the revised manuscript.
> >
> > > Q3: Can you elaborate on the training protocols and hyperparameters used (e.g., learning rates, batch sizes, number of epochs, and computational resources)?
> >
> > We added as much information as possible in the revised version within the page limit and included additional details in the Appendix Section B of the revised manuscript. However, we are unable to share the source code, as certain components of the algorithm are implemented in a commercial product, but we are committed to offering further information upon request.
> >
> > > Q4: Are there external validation or additional experiments to demonstrate the generalizability of the model?
> >
> > As mentioned in our response to the weaknesses section, we have expanded Table 1 from the original submission to further demonstrate the model's generalizability. Specifically, we now present case-level performance for single-year MRI scans across three additional datasets: the publicly available ProstateX-Challenge dataset and two private datasets, all of which include PI-RADS scores and biopsy results (with PI-RADS scores for the ProstateX dataset assigned by our radiology expert). Since these datasets originate from different cohorts and institutions than the training set, they serve as external validation for our model. We added these descriptions in Section 3.3 and detailed results in Appendix Section A of the revised manuscript.
> >
> >
> >
> > | Model       | ProstateX   |   (N = 343)|           | Private A  |  (N = 170) |            | Private B | (N = 181)|            |
> > |---------------|--------|--------|--------|--------|--------|--------|--------|--------|--------|
> > |                  | P ≥ 3  | P ≥ 4  | G ≥ 2  | P ≥ 3  | P ≥ 4  | G ≥ 2  | P ≥ 3  | P ≥ 4  | G ≥ 2  |
> > | nnU-Net           | 0.840         | 0.856      | 0.808       | 0.858  | 0.841  | 0.790  | 0.846  | 0.883  | 0.770  |
> > | Res-UNet         | **0.873**   | 0.878      | 0.822       | 0.813  | 0.818  | 0.823  | 0.821  | 0.828  | 0.768  |
> > | Our framework | 0.861        | **0.888** | **0.832**  | **0.878** | **0.891** | **0.869** | **0.852** | **0.886** | **0.824** |
> >
> > **Note:** N represents the number of cases in the dataset, P denotes the PI-RADS scores, and G stands for the GGG scores.

---

> ### Comment · Area_Chair_XehW · 2025-03-17
> **Rebuttal Response: Reviewer namb**
>
> Dear reviewer namb,
>
> Could you please review the author's rebuttal and submit your final response for the rebuttal. It's very important to have your final review and final decision.
>
> Best, AC

---

> ### Comment · Reviewer_namb · 2025-03-17
>
> I have submitted my final response before the deadline. Let me know if I missed anything. The comments I see are given as follows, where the final rating and the justification are already displayed:
>
> Preliminary Rating: 4: Weak accept
>
> Final Rating: 4: Weak accept
>
> Justification Of The Preliminary Rating:
> The authors proposed a multi-scale foundation model. The manuscript provides both lesion-level and case-level performance metrics and includes ablation studies. By comparing the model’s performance with radiologists and a state-of-the-art CNN-based baseline, the work demonstrates its potential on clinical decision-making.
>
> Justification Of The Final Rating:
> Thank you for addressing some of my earlier concerns with additional evidence.
> I have maintained my original rating.

---

### Author Rebuttal · Authors · 2025-03-08

**Rebuttal:**

We sincerely thank all the reviewers for your thorough review and the time you dedicated to understanding and recognizing our work's contribution. We also appreciate your insightful comments, which have been invaluable in enhancing the quality of our manuscript.

Based on the rebuttal policy, we have revised our manuscript based on your comments, adding as much detailed information and experimental results as possible within the page limit and including additional information in the appendix.

**Supporting Material:**

/attachment/4bd91c4dcff04200a7d6780266cb401f6b112cee.pdf

---

### Meta-Review · Area_Chair_XehW · 2025-03-21

**Recommendation:** Accept (Poster)
**Confidence:** 5

**Metareview:**

This paper presents a clinically relevant deep learning framework for predicting prostate cancer progression using longitudinal bpMRI scans. The reviewers acknowledged the paper's strengths, including the multi-scale foundation model, comprehensive evaluation metrics, and clinical significance. While initial concerns were raised regarding generalizability, reproducibility, and methodological clarity, the authors effectively addressed these points in their rebuttal and revised manuscript. Specifically, the addition of external validation results across multiple datasets significantly strengthens the paper's claims of generalizability. The authors also provided detailed explanations regarding synthetic lesion pair generation, threshold selection, training parameters, and annotation details, enhancing the paper's clarity and reproducibility. Furthermore, the inclusion of a baseline comparison with nnU-Net addressed concerns about sufficient benchmarking.

Despite some limitations, such as the inherent challenges of acquiring multi-institutional longitudinal data and the lack of full code availability due to commercial components, the paper demonstrates a valuable contribution to the field. The authors' efforts in addressing reviewer concerns and enhancing the paper's rigor warrant acceptance. Therefore, I recommend accepting this paper, recognizing its clinical relevance and potential impact. Future work should focus on continuing to seek and incorporate additional external validation datasets, exploring ways to improve reproducibility, and further investigating the impact of different training parameters and threshold selection.